# Target Identification Using Homopharma and Network-Based Methods for Predicting Compounds Against Dengue Virus-Infected Cells

**DOI:** 10.3390/molecules25081883

**Published:** 2020-04-18

**Authors:** Kowit Hengphasatporn, Kitiporn Plaimas, Apichat Suratanee, Peemapat Wongsriphisant, Jinn-Moon Yang, Yasuteru Shigeta, Warinthorn Chavasiri, Siwaporn Boonyasuppayakorn, Thanyada Rungrotmongkol

**Affiliations:** 1Center for Computational Sciences, University of Tsukuba, 1-1-1 Tennodai, Tsukuba, Ibaraki 305-8577, Japan; kowith@ccs.tsukuba.ac.jp (K.H.); shigeta@ccs.tsukuba.ac.jp (Y.S.); 2Program in Bioinformatics and Computational Biology, Graduate School, Chulalongkorn University, Bangkok 10330, Thailand; kitiporn.p@chula.ac.th; 3Advanced Virtual and Intelligent Computing (AVIC) Center, Department of Mathematics and Computer Science, Faculty of Science, Chulalongkorn University, Bangkok 10330, Thailand; Peemapat.W@gmail.com; 4Department of Mathematics, Faculty of Applied Science, King Mongkut’s University of Technology North Bangkok, Bangkok 10800, Thailand; apichat.s@sci.kmutnb.ac.th; 5Institute of Bioinformatics and Systems Biology, National Chiao Tung University, Hsinchu 300, Taiwan; moon@faculty.nctu.edu.tw; 6Department of Biological Science and Technology, College of Biological Science and Technology, National Chiao Tung University, Hsinchu 300, Taiwan; 7Center for Intelligent Drug Systems and Smart Bio-devices, National Chiao Tung University, Hsinchu 300, Taiwan; 8Department of Chemistry, Faculty of Science, Chulalongkorn University, Bangkok 10330, Thailand; warinthorn.c@chula.ac.th; 9Applied Medical Virology Research Unit, Department of Microbiology, Faculty of Medicine, Chulalongkorn University, Bangkok 10330, Thailand; siwaporn.b@chula.ac.th; 10Biocatalyst and Environmental Biotechnology Research Unit, Department of Biochemistry, Faculty of Science, Chulalongkorn University, Bangkok 10330, Thailand

**Keywords:** dengue, homopharma, network-based analysis, phenolic lipid, target identification, bioinformatic, virus-host interactions

## Abstract

Drug target prediction is an important method for drug discovery and design, can disclose the potential inhibitory effect of active compounds, and is particularly relevant to many diseases that have the potential to kill, such as dengue, but lack any healing agent. An antiviral drug is urgently required for dengue treatment. Some potential antiviral agents are still in the process of drug discovery, but the development of more effective active molecules is in critical demand. Herein, we aimed to provide an efficient technique for target prediction using homopharma and network-based methods, which is reliable and expeditious to hunt for the possible human targets of three phenolic lipids (anarcardic acid, cardol, and cardanol) related to dengue viral (DENV) infection as a case study. Using several databases, the similarity search and network-based analyses were applied on the three phenolic lipids resulting in the identification of seven possible targets as follows. Based on protein annotation, three phenolic lipids may interrupt or disturb the human proteins, namely KAT5, GAPDH, ACTB, and HSP90AA1, whose biological functions have been previously reported to be involved with viruses in the family Flaviviridae. In addition, these phenolic lipids might inhibit the mechanism of the viral proteins: NS3, NS5, and E proteins. The DENV and human proteins obtained from this study could be potential targets for further molecular optimization on compounds with a phenolic lipid core structure in anti-dengue drug discovery. As such, this pipeline could be a valuable tool to identify possible targets of active compounds.

## 1. Introduction

Dengue virus (DENV) belongs to the flavivirus genus, which is normally transmitted by mosquitoes [1,2,3,4]. This virus causes dengue fever, which is a major health problem worldwide, especially in tropical and subtropical areas [2,3,4,5]. Over 25 million people are now at risk of this infection every year that affects world economies [6,7]. This infection affects not only local people, but also travelers who visit high-risk regions [8]. The major fatal cases are caused by the mechanism of antibody-dependent enhancement during the secondary infection by different serotypes [9,10,11].

Although a vaccine for DENV has been launched in the last few years, its effectiveness and safety are still ambiguous [12,13]. Nowadays, the procedure to cure DENV infections is symptomatic treatment, since there is neither any specific antiviral drug nor any efficient prevention against DENV infection. Many potent compounds are in the process of drug discovery and design [14,15,16,17,18]. These compounds can directly interrupt the viral structural and enzymatic proteins or disrupt the associated human genes and/or proteins in the cell. However, the mechanism or target of some candidate inhibitors remains unclear and there is still the need for more studies to support and confirm their inhibitory effect.

The liquid from a cashew nutshell (*Anacardium occidentale*) consists of various phenolic lipid compounds, including anacardic acid, cardol, and cardanol. Previous reports suggested that their biological activities include anti-oxidant, anti-cancer, anti-inflammation, anti-microbial and anti-viral properties [19,20,21,22,23]. Mixed anacardic acid was demonstrated to have potential compound to inhibit Hepatitis C virus (HCV) life cycle [23], and may act through perturbation of the host target at lysine acetyltransferase 5 (KAT5 or TIP60), which is involved in viral replication [19,20]. Recently, cardol triene was reported to inhibit all dengue serotypes, and molecular dynamics simulation and cell-based functional assays suggested the potential target of this compound is the kl-loop of the DENV envelope (E) protein [21]. According to these data, the phenolic compound might interfere and inhibit many targets on viral and human proteins as a multi-target inhibitor. Therefore, not only viral proteins but also human factors could also be interrupted by these active compounds, raising the possibility of adverse side effects. Although many studies have revealed the ability of potent compounds that could interrupt several proteins and host factors in human cells [7,14,15,24,25,26], one of the challenging quests is to search for the probable protein targets of active compounds, which is an arduous step for drug discovery and design [27,28,29,30,31]. Once discovered, the potent molecules can be further optimized to increase their efficiency using the information on the interaction between the compound and key residues in the target binding site. To identify the target of known compounds, molecular docking can be used to predict the possible target of the given molecule by comparing the binding affinity and molecular posse in the binding site. However, the limitation of this method is the availability of known protein structures as targets [32,33]. In the case of no protein target available, the chemogenomics approach can be applied to determine the relationship between the chemical molecule and biological target using biological bigdata analysis [34].

Molecular information on the interaction between known compounds and their protein targets can be determined as a template in order to search for the novel potent inhibitors using a structure-based virtual screening technique [35]. Using the homopharma concept, the topology and chemical functional group of an active compound that is similar to an active molecule would bind to the same target protein at the specific binding site. Previous studies have revealed the advantage and success of this technique in the search for new inhibiting mechanisms of known drugs for drug repurposing [36]. Moreover, there is also the ability to use the approach to predict the target of an active compound, which is one of the most critical steps to bridge the gap between experimental and computational studies [37].

Network-based analysis is an important tool in biomedical researches and has been applied in several studies of molecular mechanisms. This technique can analyze the relationship among huge datasets in many fields. The biological applications of this method have become the powerful tools for in silico studies, such as repositioning and repurposing existing drugs to treat new diseases, evaluating protein–protein interactions, providing and exploring the biological pathways, or identifying the targets of active molecules. The interactions between compounds and targets can be represented by edges inferred from several existing databases [38]. However, one of the limitations of this technique is the bottleneck effect of information, where data are not yet available in the database [39]. In addition, expanding the scope of the initial information such as the network-based interaction between drugs and targets or the profiling of hit compounds from chemical screening, can increase the possibility of success [40]. The DENV–human interactome network is the biological network that describes the interaction between viral proteins and human host factors according to their interaction database [41]. Many topological parameters are required for identifying the importance of each node in the network, especially the degree, clustering coefficient, and centrality. For the network of drug target identification, the betweenness centrality (BC) measurement is widely used to measure the load of the shortest paths passing through a node of interest [42], where the significant node can be represented by a high BC value.

In this study, we aimed to seek possible targets in both the DENV and host proteins of the three known bioactive phenolic lipid compounds, namely anacardic acid, cardol, and cardanol, in order to describe the inhibiting mechanism against DENV using a combined strategy of the homopharma concept and network-based analysis. We derived 40 protein targets related to the 37 active compounds from the Bioassay database. The result suggested that KAT5 and its envelope proteins may serve as the potential targets of these three phenolic lipids. The pipeline, summarized in Figure 1, could be beneficial for predicting targets in order to develop a new drug against DENV in further studies.

## 2. Results and Discussion

### 2.1. Similar Compounds of Phenolic Lipids for Predicting Target

We explored structurally similar compounds to the given three phenolic lipids (anacardic acid, cardol, and cardanol) from the PubChem database. The topologies of the phenolic lipids were translated into binary codes and the similarity index between their codes and other compounds were computed. Each compound was labeled by a compound identification number, as shown in Figure 2. In this step, 981 compounds with more than 95% similarity to the three phenolic lipids were found using the fingerprint Tanimoto-based two-dimensional (2D) similarity search (Table 1). Nearly 90% of compounds in this database have not been evaluated in any experimental test for bioactivity [43]. Thus, we selected only 37 similar compounds that were confirmed as an active compound in at least one bioassay for our further analysis.

To measure the pairwise compound similarities, hierarchical clustering was conducted to define and group the phenolic lipids and the 37 structurally similarly active compounds. For each pair of compounds, the similarity was measured by the Pearson correlation and translated to distance values. The data was visualized as a dendrogram and heatmap matrix (Figure 2). All the active, similar compounds and phenolic lipids could be classified into three clusters; herein designated as CI (yellow), CII (blue), and CIII (pink), based on the agglomerative hierarchical cluster algorithm using a single linkage method. Anacardic acid, cardol, cardanol, and 20 active similar compounds were grouped in the CI, while CII and CIII consisted of 10 and seven active similar compounds, respectively.

By considering the topology similarity, similar compounds in each cluster were used to search for potential target proteins using the bioassay analysis in the PubChem database. In total, 13, 16, and 19 protein targets were found for CI, CII, and CIII, respectively. Only one enzyme, arachidonate 5-lipoxygenase, was a common target for all three clusters of active similar compounds. The CI and CII had six common protein targets, which were serine/threonine kinase 33, glyceraldehyde-3-phosphate dehydrogenase, arachidonate 5-lipoxygenase, and three histone acetyltransferases (p300, KAT2B, and KAT8).

### 2.2. Network-Based Construction and Centrality Analysis

By combining the DENV proteins from the DENV-human interactome (DenvInt) database and human proteins from the STRING protein-protein interaction (PPI) database, the extended DENV-human interaction network (DenvIntS) was constructed, as shown in Figure 3. This network consisted of 478 human proteins and 10 dengue proteins (C, E, NS1, NS2A, NS2B, NS3, NS4A, NS4B, NS5, and PrM/M). The whole network followed the power-law distribution and the properties of the network are represented in Table 2. The centrality scores calculated from each node in this network are shown in Appendix A. The average degree of connections of the DENV proteins interacting with the human proteins was 67.6, which was much higher than the average degree of the whole network. This indicated that the viral proteins intervene in the human proteins to survive and maintain their life as expected. A comparison of the average node properties between the human and the DENV protein nodes is shown in Table 3. All centrality measures (degree, eigencentrality, BC, and closeness centrality (CC)) demonstrated higher values in the DENV protein nodes. The dengue proteins showed a hub property since they had high degree but low clustering coefficient in average. Their neighboring proteins were not much connected as the average value of the clustering coefficients was quite low compared to those of the all protein in the network and also to those of the only human proteins. Interestingly, the number of involved triangles for the DENV proteins was much higher than the others, which suggests that most of the human proteins that connected to the DENV proteins were also connected to other proteins as well as to their own proteins.

### 2.3. Prediction of DENV Related Target

In previous studies, certain compounds were found to interact with many proteins, like a multi-target of an inhibitor [44,45,46]. Therefore, we searched for the possible targets of both the DENV protein and human factors by discovering in-depth information in the database. From compound to target, a set of 37 bioactive structurally similar compounds linked to the PubChem bioassay database were obtained (Section 2.2). Although up to 403 bioassays were related to these compounds, only 163 bioassays revealed effective bioactivity and 92 assays confirmed the existence of their target. From these, the redundant data were removed leaving only 40 possible targets relevant to the 37 active similar compounds.

Herein, we sought to identify possible targets of the DENV inhibitor. Therefore, a network-based screening technique was used to refine the results (Figure 4). From a total of 488 components of the DenvIntS PPI network and 40 targets of 37 active similar compounds, only intersect data were considered as our possible targets that were related with the DENV infection (Figure 4A and details in Appendix A). The result indicated that KAT5 and GAPDH might be potential targets of the three phenolic lipids based on the homopharma concept. These targets were enlarged and used to construct a sub-network to find the associated neighboring proteins within the DenvIntS PPI network and to construct a new sub-network (Figure 4B). This sub-network could refer to the relationship between the phenolic lipids and their targets. These compounds were expected to interfere with viral and human proteins in the endoplasmic reticulum (ER)-related translation and ER-Golgi transportation pathways [23]. For KAT5, it was linked to a viral protein and 10 human factors, while GAPDH node connects to two DENV and seven human proteins, while both KAT5 and GADPH are related to the common human node, ACTB, that plays several roles in the human cells.

To understand the inter-relationship between the predicted targets and the other human proteins by searching through the bioassay database, the chord diagram in Figure 5 shows two different levels of protein data obtained from the network-based screening technique (node) and bioassay analysis (circular segment). Direct DENV-human interactions are represented as a black line. The referred DENV-human and human-human association is illustrated as grey lines and background color (yellow and pink), respectively. It can be seen that KAT5 is related to the DENV NS5 protein and 19 other human proteins with DNA repair and apoptosis functions. Interestingly, KAT5 overexpression was related to cell proliferation [47,48] and KAT5 knockout induced apoptosis [49]. It is also known that DENV induces cellular apoptosis and so KAT5 could be one of the downregulated proteins triggering apoptosis in DENV infected cells.

For GAPDH, it directly interacts with two DENV proteins (NS1 and E) and 28 other human proteins. The GAPDH protein is responsible for many cellular pathways such as glycolysis, membrane fusion, microtubule bundling, phosphotransferase activity, nuclear RNA export, DNA replication, and DNA repair. The DENV NS1 and E proteins could interact with GAPDH during fusion and transport, respectively. Among 45 unique nodes, not only ACTB, but also HSP90AA1 is also the common interacting proteins with KAT5 and GAPDH. The ACTB and HSP90AA1 associated with NS3 and E viral protein, respectively (see Figure 5). All possible human targets (KAT5, GAPDH, ACTB, and HSP90AA1) should undergo further functional analysis that is relevant to DENV infection using annotation techniques.

Additionally, we also searched for the protein target of the three phenolic lipids (anacardic acid, cardol, and cardanol) using the similarity ensemble approach (SEA) [50] applied to PubChem database in order to compare with our protocol. Only anacardic acid was found in this database with the maximum Tanimoto coefficient (maxTC) = 1.0) with the three associated targets, SUMO-activating enzyme subunit 1 and two histone acetyltransferases (p300 and KAT2B). The results are in good agreement with the network of active-interacted proteins (Figure 4A and Appendix A). However, these proteins have not been relevant to DENV infection according to the DenvIntS PPI network (yellow area). Interestingly, with the decrease of the structural similar criteria of query to 50% using the SEA tool, the two possible proteins (KAT5 and GAPDH in green area) related to DENV infection were revealed. This indicates the reliability of our method that solidifies enough to proceed and identify the possible target of the active inhibitors.

### 2.4. Viral Target for Phenolic Lipid Compounds

In the DENV life cycle, the virus enters the human body via a mosquito bite and then infects nearby dendritic cells using specific cell receptors, followed by receptor-mediated endocytosis and fusion using the DENV E protein conformational change. This protein is activated by the acidic conditions in the endosome, releasing the nucleocapsid into the cytoplasm. Uncoating and translation are initiated at the ER membrane, which has attached ribosomes. After that, the synthesized single polyprotein is cleaved by the NS2B/NS3 protease. Viral RNA is replicated using a replication complex consisting of several viral and host proteins. Finally, the virus is assembled in the ER lumen and the new virions are transported through the ER–Golgi complex before budding out of the host cell [51,52].

According to the DenvIntS network (Figure 4), we found that the possible DENV protein targets were the E, NS1, and NS5 proteins. The obtained results are somewhat supported by the previous study using time-of-drug-addition (TOA) in which a series of phenolic compounds could interact with multiple targets at both early and late stages of the viral life cycle [21]. Since these proteins contribute to several roles in the viral life cycle, they are important targets for developing flavivirus inhibitors [18,35,53]. The NS1 protein is involved in several functions throughout the viral life cycle and immune evasion and can be used for DENV detection. Some studies have reported that the NS1 protein can serve as a drug target for DENV [54,55]. However, although the binding region of this protein has been predicted using computational tools, this has not been experimentally confirmed. For the most conserved DENV protein NS5, the SAM binding pocket is the methylation site for flavivirus, which is widely used in virtual screening and drug design [18,56,57]. Though the kl loop or βOG pocket located between the domains 1 and 2 on the DENV E protein has been confirmed to have an inhibitory effect against DENV and other flaviviruses using experimental and computational assays [18,21,58,59], other biding sites for antibody design, and compound screening have been reported on this structural protein [60,61,62,63,64,65].

### 2.5. Human Protein Target Annotation

The online databases were used to annotate the 40 predicted targets according to the previous step (Section 2.4). Gene enrichment analysis revealed five molecular functions (transcription regulator activity, binding, structural molecule activity, catalytic activity, and transporter activity), and 10 biological processes which were mainly involved in cellular and metabolic processes. The details are shown in Figure 6.

The four possible human targets of these phenolic lipids have been described for their biological functions. KAT5 is part of the Tip60-EP400 acetylation complex that interacts with the DENV NS3 and NS5 proteins and has several functions, such as transcriptional regulation, protein acetylation, induction of apoptosis, histone complex formation, and chromatin remodeling process. Moreover, it has been reported to have an antiviral role in infection with HIV and some flaviviruses [19,66]. Recently, a KAT5 inhibitor was shown to have an inhibitory effect against the Chikungunya virus, at the step of viral replication with a therapeutic index of 4.65 [19], and against HCV [20,23].

The GAPDH enzyme is involved in several cellular mechanisms and plays a role in the glycolysis pathways and nuclear activities, including apoptosis induction. It can be complexed with the NS5 RdRp protein in Japanese Encephalitis (JEV) infected cells [67] and NS3 in DENV infected cells [68]. The interaction between NS3 and GAPDH resulted in a decreased glycolytic activity. This protein shows a crucial function in the early steps of viral attachment, entry, and fusion of HCV and DENV infection, which has been confirmed by siRNA-mediated GAPDH silencing [69]. However, GAPDH does not directly bind to the NS5 or NS3 proteins of JEV but it interferes with viral RNA replication by interacting with RNA [67,70]. In the case of DENV, GAPDH has been revealed to directly interact with DENV NS3 by disturbing the helicase domain. The NS3-GAPDH interaction elevates the NS3 ATPase activity and decreases the GAPDH glycolytic activity in hepatic cells [68]. Additionally, another benefit of GAPDH is as an antigen for use as a control in Western blots [71].

The key function of ACTB is actin production for cell motility and contraction, structure, integrity, and intercellular signaling. The interference of ACTB or actin motion can inhibit DENV at several steps of viral entry, assembly, and maturation [72]. Furthermore, it has been reported that ACTB can bind to the TAT protein of HIV, resulting in an increased efficiency of viral replication [41]. Both ACTB and GADPH have been reported to play a role in the stability in dengue fever patients [73], and can used as potential molecular markers for DENV prediction.

For HSP90AA1, the subunit of heat shock protein (HSP90) that is conserved among flavivirus, it has been shown to play a pro-viral role in the replication cycle of several viruses, predominantly through stabilization of specific viral proteins. Inhibitors of Hsp90 can block the replication of many viruses. For DENV, the HSP90 inhibitor showed the ability to decrease the level of the DENV E protein in host cells [74]. Previous studies showed the inhibition mechanism of the focused phenolic lipids was mainly at the early stage of viral infection [21]. Therefore, the viral E protein and/or human HSP90 might be disturbed by these compounds. Overall, KAT5, GAPDH, ACTB, and HSP90AA1 are possible human targets of these compounds, based on the previous study and several steps of target identification in this work.

## 3. Materials and Methods

### 3.1. Data Collection

It was previously shown that the three phenolic lipid compounds (anacardic acid, cardol, and cardanol) were active inhibitors against DENV infected cells. The 2D chemical structure of these molecules in the SMILES format were used to search the PubChem database (https://pubchem.ncbi.nlm.nih.gov/) [75] using the fingerprint Tanimoto-based 2D similarity search. The compound structures were separated into substructures and the structural information was encoded into a binary string. Structural similarity between active compounds and compounds in the databases was evaluated by comparing the molecular fingerprint and scoring using the Tanimoto algorithm (1) [76,77]. The similarity coefficient (Tanimoto) is the ratio between the number of common molecular fingerprints ($N_{AB}$) and the different numbers of each set of molecular fingerprints ($N_{A}+N_{B}-N_{AB}$) [76,78], as shown in Equation (1):(1)$$Tanimoto=\frac{N_{AB}}{N_{A}+N_{B}-N_{AB}}$$

In this case, the similarity search of the phenolic lipid compounds in the PubChem database collected only molecules whose similarity scores with the respective query molecule were greater than or equal to a 95% Tanimoto threshold. To narrow the search, only compounds with a reported bioactivity were selected as representative similar compounds. The sets of active similar compounds and queries were then clustered using ChemMine tools [79] based on pairwise compound similarities.

### 3.2. Related Target Evaluation

The set of active similar compounds was used to predict the relevant target by searching from the bioassay databases that contain an extensive and diverse array of information about biological interactions [80]. This database collects the information from several data sources, including ChEMBL [81], TOX21 [82], BindingDB [83], and NCGC [84]. Over a hundred million data exist in this database, consisting of small-molecule and RNAi screening data using several kinds of experimental assays that have been retrieved from many data sources. The bioactivity of each compound was classified for its efficiency as “active”, “inactive”, and “inconclusive” based on the bioassays data [85]. Both gene and protein can be the target of similar compounds. Therefore, the targets were collected and grouped by the type of data. Only compounds reported as “active” in each experimental assay and showed their target were selected and used in this study. The validity of this method was confirmed by the similarity ensemble approach using the SEA search tool (http://sea.bkslab.org) [50].

### 3.3. Network-Based Construction and Analysis

In this paper, the main interactome was focused on the DENV–human protein–protein interactions downloaded from the DENV protein interaction database (DenvInt). The network was constructed based on several assays, such as a high-throughput yeast two-hybrid, siRNA, complex pulldowns, and Western blot methods [41]. Data in this interactome is annually updated and compared to the other available interactomes. Basically, the nodes of the network represented both DENV proteins and the human factors. In between them, there were a set of connecting lines (edge) to show the relationship among nodes as an undirected and unweighted network. The scope of interactions in the original interactome was expanded by merging the associated proteins from the STRING protein-protein interaction network database to be the extended network (DenvIntS) [86]. Then, this DenvIntS PPI network was calculated and visualized using the Gephi software [87] for various node property measures, such as the degree of edge, eigencentrality, CC, BC, clustering coefficient, and number of triangles, so as to characterize the importance of each node.

The degree is a measure of the number of connections for a certain node in the network. It is widely used to identify hub nodes in a network, if the node has more connections to other nodes. This measure is well-known to qualify how important a node is related to the essentiality of genes in bacteria, yeast, and humans [88,89]. Eigencentrality is a centrality measure based on the calculation of the eigenvector of an adjacent matrix under the assumption that any node and their neighboring nodes have more or less similar scores of importance level in the network. Nodes that have high connections (high degree) should have higher scores as well as their neighbors also have high scores. The formal calculation can be found in [88,89]. The CC is a centrality measure of how far or close is a node, if walking from the other nodes to it. The calculation is based on the inversion of the shortest path distance of this node to the other nodes [42,89], as shown in Equation (2):(2)$$CC\left(k\right)=\overset{N}{\underset{i=1}{\sum }}\frac{1}{d\left(k,i\right)}$$
where d(k,i) is the distance between node k and node i. The CC increases when the distance to another node decreases.

The BC is mostly used to describe the importance of each node in the network by calculating all the shortest paths between the two other nodes. In a biological network, the BC is more important than the other topology analyses [89,90,91]. The node with many short paths will show a high BC and indicate the essentiality of this node. The BC of node k (BC (k)) can be computed by the sum of the total number of shortest paths ($\sigma_{ij}$) from node i to node j, as shown in Equation (3):(3)$$BC\;\left(k\right)=\sum_{i\neq j\neq k}^{}\frac{\sigma_{ij}\left(k\right)}{\sigma_{ij}}$$

The clustering coefficient is a measure of the local density of a node by comparing the true connections among the neighbors of the node to all possible connections among the neighbors [89,92], and ranges in value from 0 to 1 as the probability of getting a connection among neighbors. If all the neighbors are connected, the clustering coefficient is 1. If all the neighbors are not connected, the clustering coefficient is 0. The number of triangles is a measure of how many triangle forms the node is involved in [93]. Triangles have been found to be a way to identify a small but important network motif, where each node in a triangle can control each other in a directed network as well as reach each other in an undirected network.

### 3.4. Identification of Targets Aassociated with DENV

The related targets from the previous step (Section 3.3) are not specific to DENV infection. Therefore, the network-based screening techniques will be used to scope down the possible targets. The interactome network developed in the previous step was used as a sieve to screen all the similar targets of active compounds, in order to search for the targets related to DENV infection. The overlapped targets were chosen and used to construct the sub-network of neighbor nodes to explore the relationship between DENV proteins and human factors using the Gephi software [87]. Subsequently, the sub-network was performed to reveal the new interaction between the DENV related protein and neighbor factor that is defined as the second level of interacted protein.

### 3.5. Target Annotation

To identify the related protein and human factor, the UniProt database (http://www.uniprot.org/) and the Human Gene Database (https://www.genecards.org) were used in this step. The UniProt database is a hub for protein information that contains a large data from sequences to proteomic level [94]. The biological process and molecular function of the potential targets were annotated using the GeneCards database [95].

## 4. Conclusions

Compounds similar to active inhibitors may bind to the same protein with similar interactions. This homopharma concept could lead to the application of databases of integrating drugs and other potential compound databases. As these databases contain huge amounts of information from many different data sources, they are often used in modern drug discovery. Infection with DENV can lead to severe hemorrhagic fever where effective treatment is yet unavailable. Previous reports suggested that these three phenolic lipids (anarcardic acid, cardol, and cardanol) inhibited all DENV serotypes at the envelope protein during fusion, as well as proteins involved in the replication complex.

In this study, we applied the homopharma concept to predict the possible protein targets for the three phenolic lipid compounds. Using various databases, almost a thousand similar structures were found from the three known active substances, based on the 2D Tanimoto similarity (95%), of which only 37 similar compounds had experimental confirmation of their effectiveness. The 40 protein targets related to these 37 active similar compounds were selected from the Bioassay database. After that, they were identified and clustered into three clusters, which were subsequently compared with DENV-host protein interactome. The human KAT5 and GAPDH proteins were identified as a related factor to DENV infection. The predicted target proteins were used to enlarge the possible target identification by constructing a sub-network using neighboring proteins from the STRING protein-protein interaction database. Functional annotation was performed to observe their biological functions. An inference approach suggested that four human proteins (KAT5, GAPDH, ACTB, and HSP90AA1) and three DENV proteins (NS3, NS5, and E protein) may serve as potential targets of these phenolic lipids. This drug target prediction technique could be beneficial for target identification in cases where the protein structure is unknown or there is a lack of supporting experimental evidence.

## Figures and Tables

**Figure 1 molecules-25-01883-f001:**
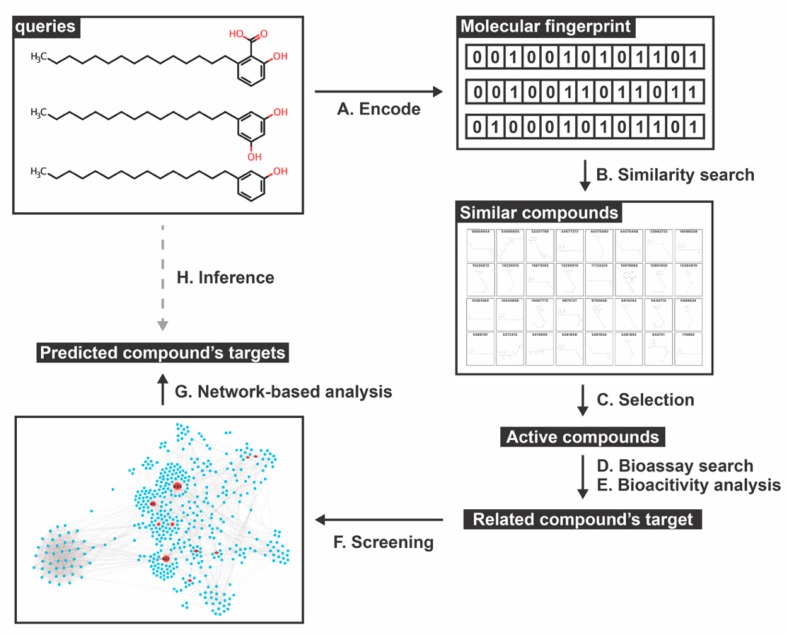
Schematic framework of this study. (A,B) Active compounds are translated into a molecular fingerprint format (binary code) as a template for the similarity search of the database based on a similar coefficient. (C–E) Among similar compounds, only the active molecules are chosen and searched for their targets using the Bioassay and Bioactivity databases. (F,G) The related compound’s targets are screened by the DENV-human interactome network (DenvInt) [41] and scored using the network-based analysis method. Finally, (H) the DENV-host related targets of active similar compounds are annotated and identified for their potential function using several databases.

**Figure 2 molecules-25-01883-f002:**
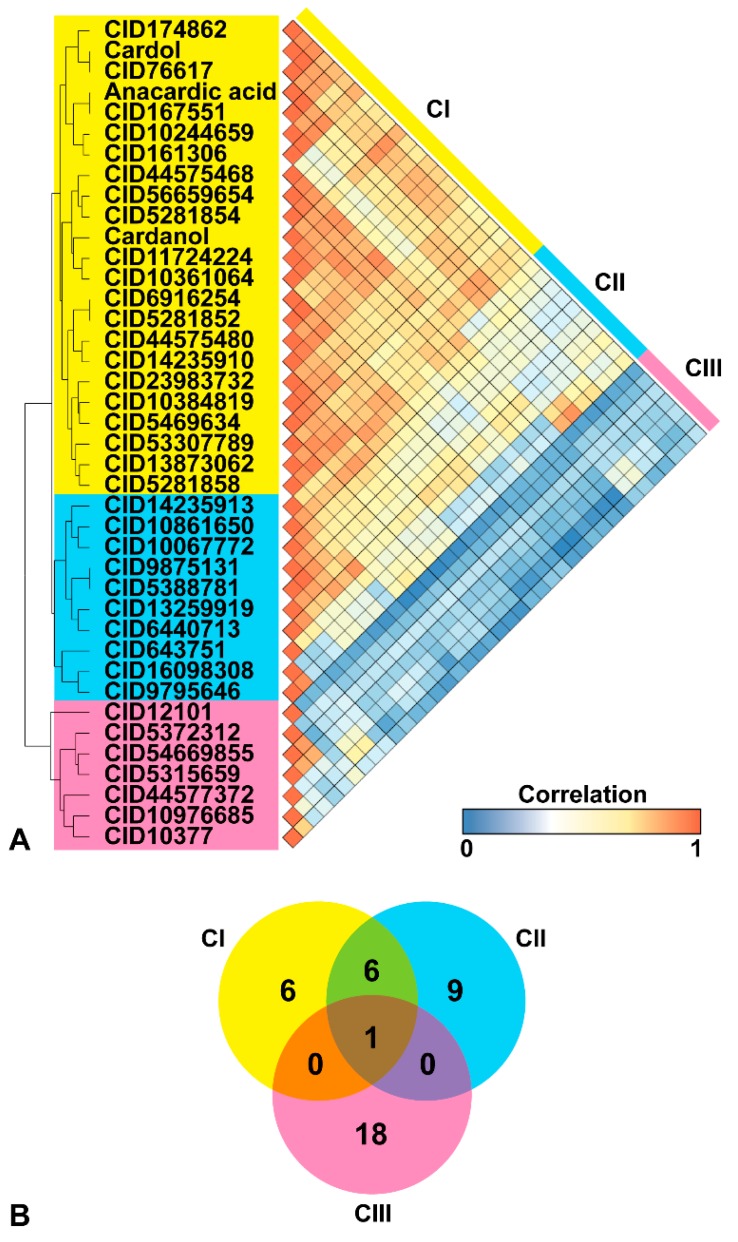
(**A**) Three clusters of phenolic lipids and active similar compounds; CI, CII, and CIII, represented as a hierarchical dendrogram (yellow, blue, and pink). The similarity score of these compounds is illustrated as a heatmap correlation that ranged from 0 (blue) to 1 (orange). (**B**) The overlap of targets from the bioassay database for the 37 active similar compounds. Only one target was a common protein of all similar compounds.

**Figure 3 molecules-25-01883-f003:**
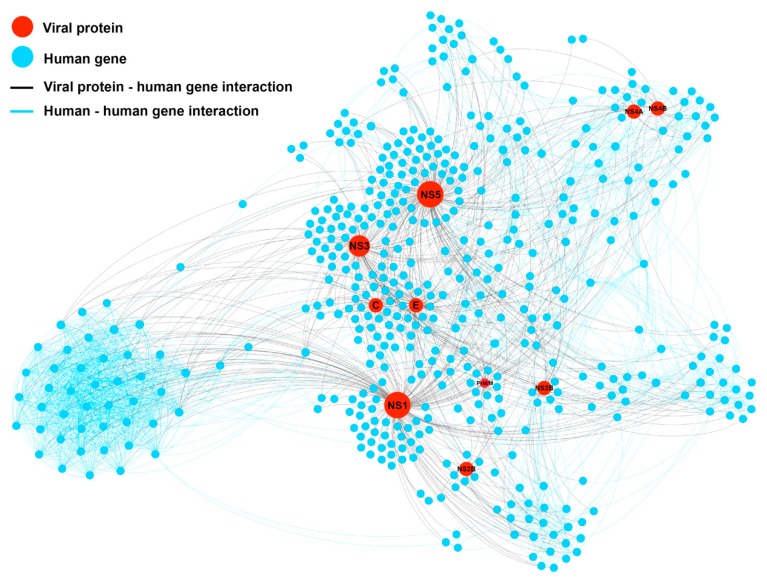
Protein-protein interaction network (DenvIntS) of DENV (red node), and the human related proteins from STRING PPI database (blue node). The interaction of DENV-human and human-human is represented as a black and blue line, respectively. The proportion of nodes is not referred to any parameters.

**Figure 4 molecules-25-01883-f004:**
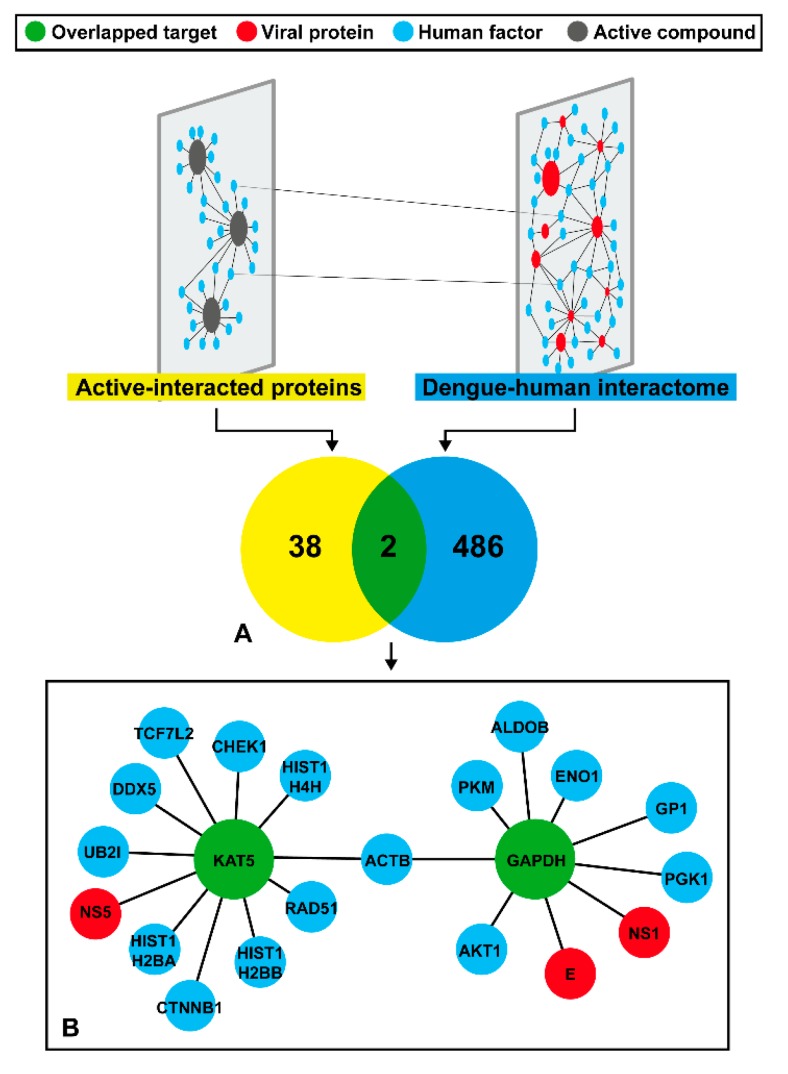
Result of the DENV-related target prediction and neighboring proteins using the network-based screening techniques. (**A**) Possible targets correlated to DENV viral infection derived from the overlap between similar compound’s targets and proteins in the DenvIntS PPI network. (**B**) The sub-network of two predicted targets (KAT5 and GAPDH) and neighboring proteins obtained from DenvIntS network.

**Figure 5 molecules-25-01883-f005:**
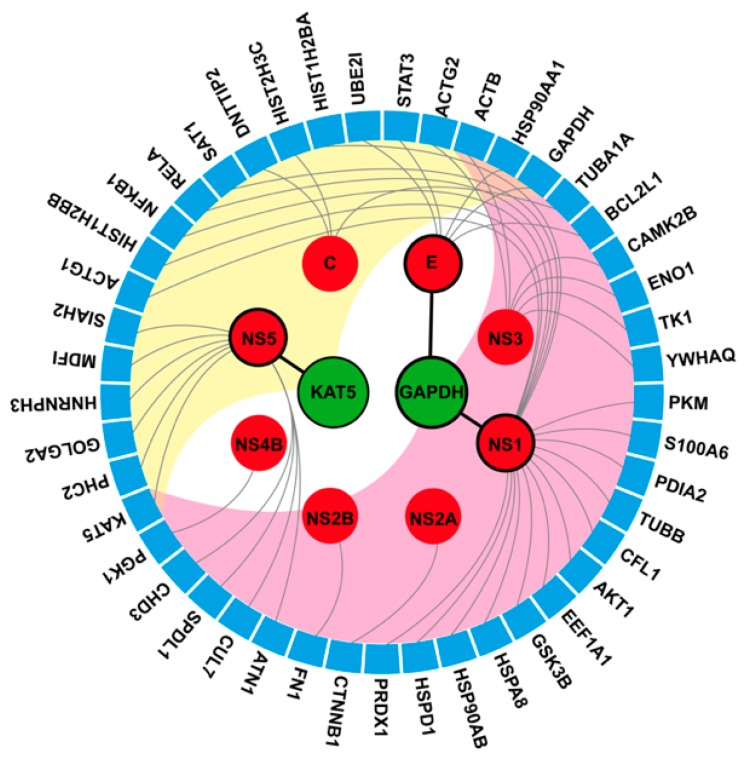
Protein-protein interaction (PPI) illustrated as a chord diagram. The predicted targets, KAT5 and GAPDH (green node), directly interact with three viral proteins (red node with black border) shown by black line. The associated viral proteins (red node without border) links to the second level of interacted proteins (blue segment) represented by the grey linkage. The connections between the predicted targets (KAT5 and GAPDH) and the second level of interacting proteins (blue segment) were defined by the pink and yellow color.

**Figure 6 molecules-25-01883-f006:**
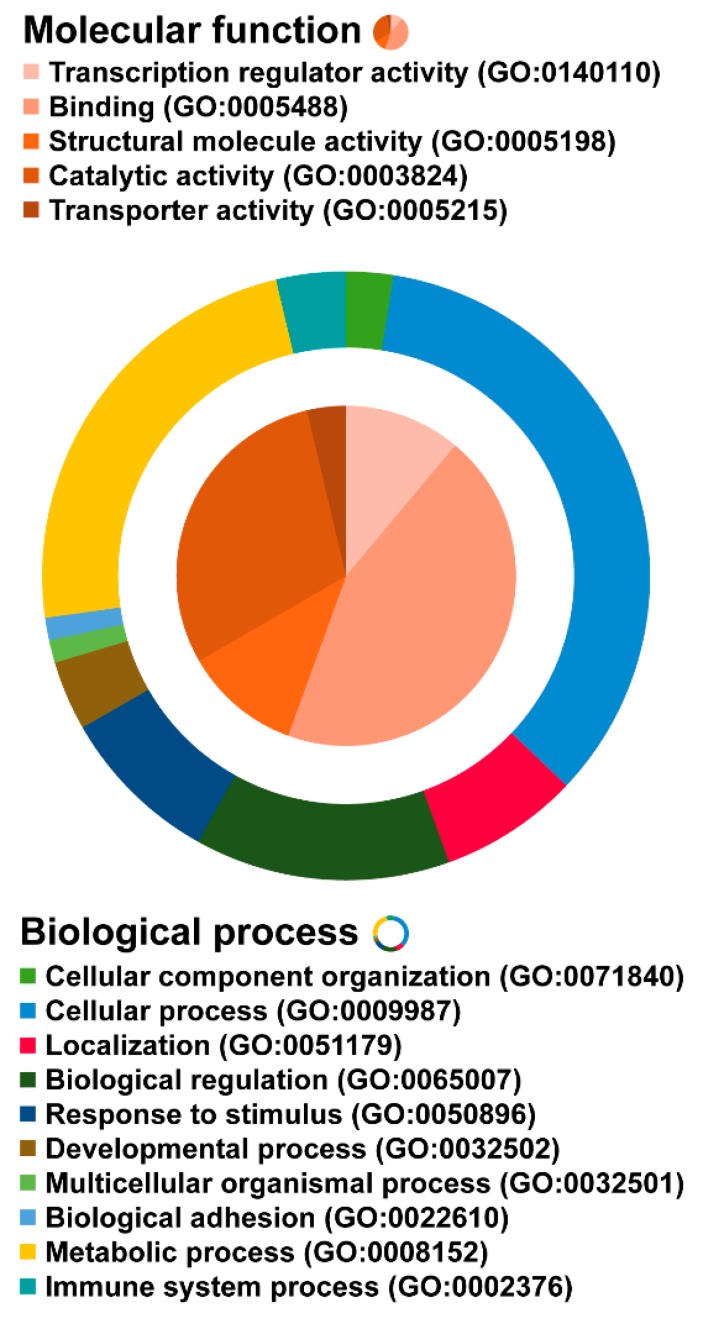
Gene oncology (GO) of the predicted targets and the other human proteins. Molecular function and biological process are represented as a pie and doughnut diagram.

**Table 1 molecules-25-01883-t001:** The number of similar compounds retrieved from PubChem database.

Phenolic Lipid Compounds	Similar Compounds
Anacardic acid	223
Cardol	311
Cardanol	447

**Table 2 molecules-25-01883-t002:** The whole network property of DENV-human protein interaction network.

Property	DenvIntS Network (Whole Network)
Nodes	488
Edges	2523
Average degree	10.340
Nodes per edge	0.193
Diameter	5
Average clustering coefficient	0.45
Average path length	2.842
Graph density	0.021

**Table 3 molecules-25-01883-t003:** Node properties for human protein nodes and DENV protein nodes.

Node Property (In Average)	All Nodes	Human Protein Nodes	DENV Protein Nodes
Degree	10.340	9.142	67.600
Eigencentrality	0.116	0.113	0.238
CC	0.357	0.356	0.410
BC	0.004	0.002	0.108
Clustering coefficient	0.315	0.320	0.054
Number of triangles	83.195	82.797	102.200

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
