# Peer review of "Target Identification Using Homopharma and Network-Based Methods for Predicting Compounds Against Dengue Virus-Infected Cells"

_molecules, 2020, doi:10.3390/molecules25081883_

Round 1

Reviewer 1 Report

The manuscript by Hengphasatporn et al. describes a network analysis of compounds and proteins in order to find targets for compounds that are proposed to have an effect on dengue disease.

The English language is quite bad, making it very hard to understand the manuscript and ideas within it.

The analysis is a bit speculative, and does not have any validation or evaluation. These two aspects are crucial to be able to replicate and value the present manuscript.

Author Response

The manuscript by Hengphasatporn et al. describes a network analysis of compounds and proteins in order to find targets for compounds that are proposed to have an effect on dengue disease. The English language is quite bad, making it very hard to understand the manuscript and ideas within it. The analysis is a bit speculative, and does not have any validation or evaluation. These two aspects are crucial to be able to replicate and value the present manuscript.

Response: First of all, we would like to thank for your beneficial comments and suggestions that allow us to improve the quality of this manuscript. The grammar mistake in this manuscript has been corrected by the native speaker (Mr. Robert Douglas John Butcher).

Undoubtedly, the experimental study will bring more insights and would help to validate the predicted targets derived from our research. However, due to the limited timing and unavailable equipment, the authors could not perform the experiments. Instead, we searched the protein targets of the focused phenolic lipids by the similarity ensemble approach (SEA), a tool for identifying the possible target of the active compound based on the similarity score of ligand topology and the biological interaction between the compound and its targets. We have revised and added the validation in the current manuscript as following text (Page 8):

“Additionally, we also searched for the protein target of the three phenolic lipids (anacardic acid, cardol, and cardanol) using the similarity ensemble approach (SEA) [50] applied to PubChem database in order to compare with our protocol. Only anacardic acid was found in this database with the maximum Tanimoto coefficient [maxTC] = 1.0) with the three associated targets, SUMO-activating enzyme subunit 1 and two histone acetyltransferases (p300 and KAT2B). The results are in good agreement with the network of active-interacted proteins (Fig. 4A and Supplemental Table S3), however, these proteins have not been relevant to DENV infection according to the DenvIntS PPI network (yellow area). Interestingly, when the decrease in the structural similar criteria of a query to 50% using the SEA tool, the two possible proteins (KAT5 and GAPDH in green area) related to DENV infection were revealed. This indicates the reliability of our method that solidifies enough to proceed and identify the possible target of the active inhibitors.”

and (Page 9):

“According to the DenvIntS network (Fig. 4), we found that the possible DENV protein targets were the E, NS1, and NS5 proteins. The obtained results are somewhat supported by the previous study using time-of-drug-addition (TOA) in which a series of phenolic compounds could interact with multiple targets at both early and late stages of the viral life cycle [21].”.

Reviewer 2 Report

The manuscript titled “Target Identification using Homopharma and Network-based Method for Predicting Compounds against Dengue Virus-Infected Cell” used homopharma and network-based method to predict drug targets in dengue and human proteins. Overall, the manuscript is well designed and executed. Review the manuscript for grammatical errors. Figure 3 is not clear and I suggest to retake figure 3.

Author Response

The manuscript titled “Target Identification using Homopharma and Network-based Method for Predicting Compounds against Dengue Virus-Infected Cell” used homopharma and network-based method to predict drug targets in dengue and human proteins. Overall, the manuscript is well designed and executed. Review the manuscript for grammatical errors. Figure 3 is not clear and I suggest to retake figure 3.

Response: Thank-you for pointing it out. Figure 3 has been improved, while the grammatical errors have been corrected in the revised manuscript.

Reviewer 3 Report

I found the manuscript very interesting. I personally do not work in exactly this area, so I’d like to see some more references to the review articles or other texts about the network based method, since there are very few, or the ones which are present only represent a narrow slice of the field. I do think the scientific part is okay.

Unfortunately, the paper is quite hard to follow, and this is partly because of poor English, or even careless language, which sometimes brings doubt if the authors read the draft (sorry). An example of such phrase is:

Mixed anacardic acid was demonstrated as potential anti-Hepatitis C virus (HCV [24]. : So anacardic acid turned into the virus?

This starts already in the abstract:

Particularly, many diseases have the potential to kill, for example, the dengue that lacks any healing agent.: The sentence literally says that diseases kill dengue.

Some inhibitors are still in the process of drug discovery: poor wording, the word drug should be removed, but then almost nothing is left, the sentence in effect means something like “The drug discovery is still taking place”  which is an obvious and non informative statement and could be removed.

The development of an active molecule is also a critical demanding for improving effectiveness.: poor wording, if you shorten the sentence it becomes “The development is also a critical demanding for improving effectiveness”. I sort of understand what was meant but it the sentence is extremely clumsy. My suggestion is to give up ornate phrases and use simple sentences, maybe even read the sentence aloud, to see if it makes sense.

Or take for example the first sentence of the introduction: “…however, it can be passed by sexual transmission in the rare case [1]” So, it was just one case of sexual transmission?

And so on, each sentence needs careful reading and reformulation.

I think the authors should clear their English first, after which perhaps the underlying science will be more easily understood.

Author Response

I found the manuscript very interesting. I personally do not work in exactly this area, so I’d like to see some more references to the review articles or other texts about the network-based method, since there are very few, or the ones which are present only represent a narrow slice of the field. I do think the scientific part is okay.

Unfortunately, the paper is quite hard to follow, and this is partly because of poor English, or even careless language, which sometimes brings doubt if the authors read the draft (sorry). An example of such phrase is:

Mixed anacardic acid was demonstrated as potential anti-Hepatitis C virus (HCV [24].: So anacardic acid turned into the virus?

This starts already in the abstract:

Particularly, many diseases have the potential to kill, for example, the dengue that lacks any healing agent.: The sentence literally says that diseases kill dengue.

Some inhibitors are still in the process of drug discovery: poor wording, the word drug should be removed, but then almost nothing is left, the sentence in effect means something like “The drug discovery is still taking place” which is an obvious and non-informative statement and could be removed.

The development of an active molecule is also a critical demanding for improving effectiveness.: poor wording, if you shorten the sentence it becomes “The development is also a critical demanding for improving effectiveness”. I sort of understand what was meant but it the sentence is extremely clumsy. My suggestion is to give up ornate phrases and use simple sentences, maybe even read the sentence aloud, to see if it makes sense.

Or take for example the first sentence of the introduction: “…however, it can be passed by sexual transmission in the rare case [1]” So, it was just one case of sexual transmission?

And so on, each sentence needs careful reading and reformulation.

Response: Thank-you for the careful review. All sentences have been corrected and the grammar mistakes in this manuscript have been proofread by the native speaker (Mr. Robert Douglas John Butcher).